# Threat Ranking to Improve Conservation Planning: An Example from the Gediz Delta, Turkey

Dilara Arslan [1,2], Kerim Çiçek [3], Ömer Döndüren [4] and Lisa Ernoul [1,*]

1  Tour du Valat, Institut de Recherche pour la Conservation des Zones Humides Méditerranéennes, Le Sambuc, 13200 Arles, France; arslan@tourduvalat.org
2  Institut Méditerranéen de Biodiversité et d'Ecologie Marine et Continentale 3(IMBE), Avignon Université, UMR CNRS IRD Aix Marseille Université, IUT Site Agroparc, BP 61207, CEDEX 09, 84911 Avignon, France
3  Zoology Section, Department of Biology, Faculty of Science, Ege University, Izmir 35100, Turkey; kerim.cicek@ege.edu.tr
4  İzmir Büyükşehir Belediyesi, Izmir 35250, Turkey; omerdonduren@izmir.bel.tr
*  Correspondence: ernoul@tourduvalat.org

**Abstract:** Mediterranean wetlands are among the most threatened natural areas. The needs and demands of an increasing human population are modifying land use and converting natural habitats into artificial areas. In order to combat these trends, effective conservation planning needs to provide clear, systematic identification of threats to find sustainable conservation strategies. In this case study, we evaluated current threats in the Gediz Delta (Turkey) using a multi-method approach. First, we did a comprehensive literature review and stakeholder interviews to identify existing threats. We then did a complete survey of the Delta through intensive fieldwork. The threats were coded and ranked using the conservation standards. We used the threat ranking and field survey to map the most vulnerable areas of the Delta. The most commonly observed threats in the field were pollution and agriculture and aquaculture activities. According to the threat ranking, the most important threats are climate change and residential and commercial development. The habitats that are most at risk are agricultural grassland habitats. The results indicate a need to extend conservation actions in the inner part of the Delta. In addition, the multi-method threat ranking approach could serve as a model to improve conservation planning in other sites worldwide.

**Keywords:** conservation planning; Gediz Delta; perceptions; threat ranking; wetlands

## 1. Introduction

The needs and demands of the increasing human population are inciting the conversion of natural lands into agricultural and urbanized areas, with significant consequences on biodiversity and human well-being [1]. Holistically protecting natural ecosystems is essential to avoid a biodiversity crisis [2]. Previous research conducted between 1997 and 2011 showed that freshwater wetlands provided the world with USD 2.7 trillion a year worth of ecosystem services [3]. However, the available data show that up to 87% of global wetland resources have been lost since the 1700s [4,5]. Unfortunately, this situation holds true in the Mediterranean basin, with wetlands being the most destroyed ecosystem in the region [4,5]. Wetlands cover 2–3% of the Mediterranean basin surface, providing critical habitats for many plant species, and breeding and feeding grounds for many animal species [6]. Wetlands host 30% of vertebrate species, more than 40% of the endemic and 36% of threatened species in the Mediterranean basin [6,7]. There has been a decline of 28% of the vertebrate population in freshwater habitats in the Mediterranean region [7]. The loss of these habitats and species means a significant loss of human well-being and biodiversity [3]. Most of these losses are due to the drainage of wetlands for residential, industrial and agricultural activities, with a high impact on coastal Delta areas [5]. Climate change is predicted to be the primary driver that will change and destroy wetlands in the

future [6,8]. Therefore, understanding the source of threats on wetlands is essential for sustainable conservation planning [5].

Conservation scientists need to identify current and potential threats in order to design effective future interventions [9]. The Conservation Standards is a framework for adaptive planning and management, based on an improved methodology that highlights objectives and goals for conservation management processes [10]. In this methodology, the International Union for Conservation of Nature (IUCN) and the Conservation Measures Partnership (CMP) designated a set of standardized classifications of direct threats to support the identification of problems and solutions in conservation management [11]. Previous research identified the main threats to wetlands as pollution (54%), biological resources use (53%), natural system modification (53%), and agriculture and aquaculture (42%); however, the presence and impacts of these threats are different from one wetland to another according to the regulation in the country and human activity [12]. Therefore, each wetland should be evaluated according to the local socio-ecological context while applying a global standard.

Gediz Delta is composed of a mosaic of salt and freshwater marshes (5000 ha), saltpans (3300 ha), and four lagoons (Homa 1824 ha; Çilazmak 725 ha; Kırdeniz 450 ha; and Taş 500 ha) (Figure 1). The Delta is recognized for its international importance for breeding and wintering waterbirds [13]. In addition, the Delta plays a vital role in maintaining the biogeographic diversity in the region. It is an important waterbird breeding site in the Mediterranean basin and hosts 80,000 wetland birds annually [13]. Furthermore, there are 20 species of fish, 35 species of amphibians and reptiles, 300 species of birds, and more than ten species of mammals and dozens of invertebrates inhabiting the Delta [13–15]. Apart from its biological importance, the Delta also provides important economic and aesthetic values [16]. Despite all these features, the Delta has faced significant land-use changes in the last 100 years, which has greatly impacted its biodiversity [17–20].

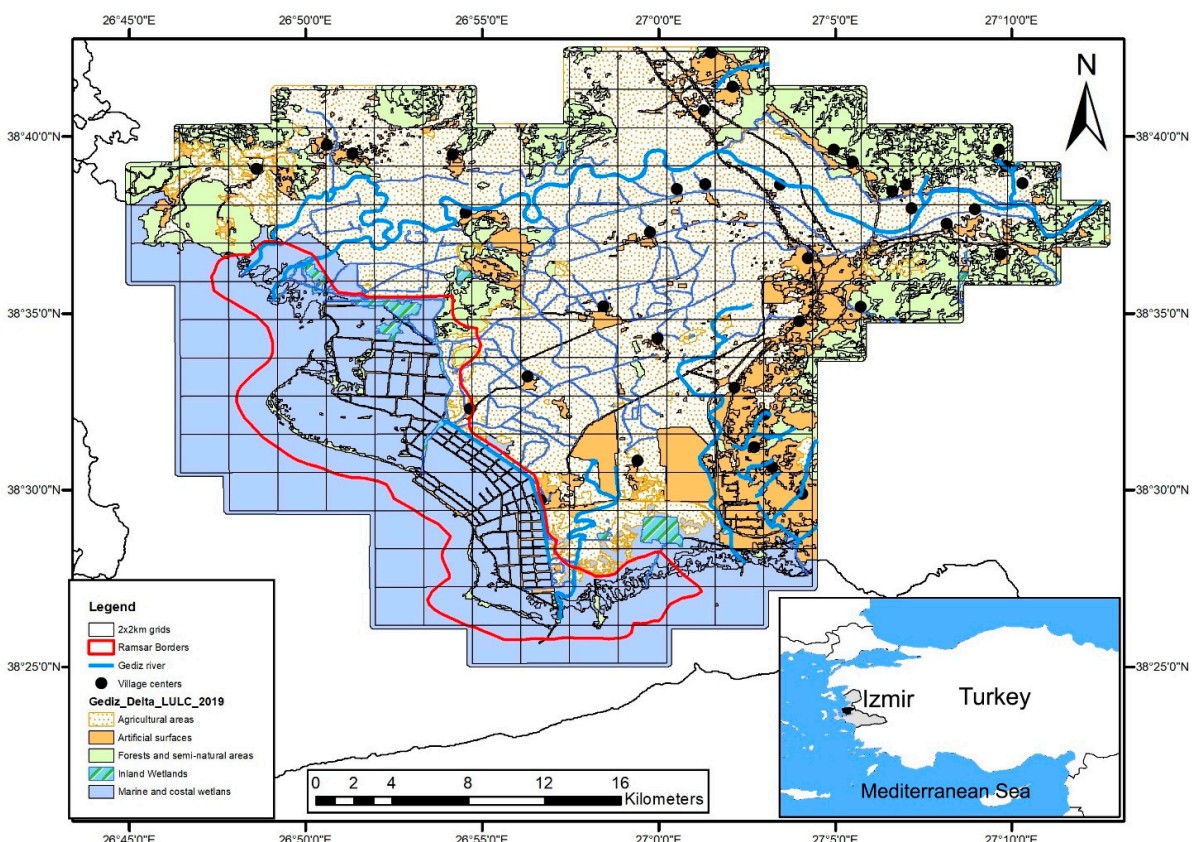

**Figure 1.** Location and principal ecosystems of the Gediz Delta in Turkey (adapted from Guelmami, 2021 unpublished data).

The Gediz Delta is located close to the city of Izmir, a metropole with a population of over 4 million people [21]. The geographic proximity to Izmir accentuates the threats and human pressures. The Delta sustains many ecosystem services for local livelihoods such as agriculture, fishing, and salt production and hosts two organized industrial zones. Despite these ecosystem services, the wetlands have been considered a threat to the local population, causing significant structural changes over the centuries [18]. This research aimed to understand the current threats in the Delta in order to improve conservation planning (e.g., suggesting concrete conservation actions to prevent further degradation or to improve the current state of the wetlands). The work was developed in 3 steps: (1) using a literature review, we gathered the threats identified in scientific journals, newspaper articles and grey literature; (2) we applied in-depth interviews to identify additional threats and gather the perceptions of threats by key stakeholders; and (3) we identified the visual threats through intensive fieldwork over 200 grids in the Delta. The information collected was coded using the IUCN threats classification system [9,22] and then ranked using the Conservation Standards methodology [11]. The threats were mapped to identify the most vulnerable zones in the Delta and provide recommendations for conservation planning. This study can be replicated in other wetlands to identify threats and improve management on a larger scale.

## 2. Materials and Methods

### 2.1. Study Site

Gediz Delta (38°30′ N, 26°55′ E) is located in the Mediterranean basin on the coast of the Aegean Sea (Figure 1). It is composed of a mosaic of freshwater and saltwater ecosystems made up of shrub forests, salt meadows, reed beds, marshes, lagoons and rivers, Salinas, and beaches [23]. The Gediz River was thought to be a flooding danger for Izmir and its course was changed 50 km to the north at the beginning of the century through a system of dykes and canals. The swamps were also drained to combat malaria [18] and, since the beginning of the 2000s, the border of the Delta was urbanized to create new residential land [17,24]. In this study, we consider the Gediz Delta to include the area between the old and new riverbeds of the Gediz River and the lower Gediz Basin.

### 2.2. Threats Assessments

The threats assessment was conducted using a three-step approach based on a literature review, in-depth interviews with stakeholders, and field visits. The information gathered using these approaches were coded using the IUCN classification system and then ranked using the Conservation Standards designed by the Conservation Measures Partnership (CMP) [11].

#### 2.2.1. Literature Review

We began our research with a literature review of both scientific journals and news articles to identify existing threats to the Delta from 1980 to 2020. Firstly, we searched the threats by Google news using the combination of the following keywords: "Gediz Deltası", "İzmir Kuş Cenneti", "Gediz Delta", "UNESCO", "Ramsar", "Izmir", and "Flamingo". Then, we searched in scientific and academic reports in Google scholar and the national thesis database (https://tez.yok.gov.tr/UlusalTezMerkezi/, accessed 11 November 2021). The search included published articles, PhD theses, management plans, and books. Only the documents containing threats specific to the Gediz Delta were used. Each threat detected in the news and literature review were classified using the IUCN threat classifications and ranked by frequency of times each threat was identified.

#### 2.2.2. In-Depth Interviews

Stakeholders from both governmental and non-governmental organizations were chosen using a targeted sampling technique [25]. We first identified key actors from the management plans of the Delta (2007 and 2019) and based on our previous experience

working in the area (Table 1). A semi-structured interview with open-ended questions was administered with those stakeholders. We structured the interviews using a conceptual chain to identify the threats by determining their direct and indirect causes [25]. The open ended-questions allowed for deeper expression of environmental problems with their different dimensions, perspectives and nuances [26], where the participants expressed their ideas without being guided by previously established responses [25]. The interviews consisted of 2 questions: (1) "What are the critical threats on biodiversity in the Delta? Please identify each threat and score them according to their importance"; and (2) "What kind of solutions should be implemented to protect nature? Please identify each solution". Before starting the interviews, the study was introduced, and participation consent was established. Interviews were conducted in Turkish, and each interview lasted between 30 and 40 min in total. The interviews were conducted from August 2019 to June 2020. The responses were first written in Turkish and then translated to English. The English translations were put into an excel spreadsheet and coded according to the IUCN threats classification. The different threat classes were then ranked according to their frequency given in the interviews. We used a $\chi^2$ test to assess the similarities and differences of threats identified by governmental and non-governmental stakeholders. The threats identified through the literature review and interviews were analyzed by clustering with "tm" and "wordcloud" packages in R [27,28].

**Table 1.** List of stakeholders in the Gediz Delta participating in the semi-structure interviews about environmental threats in the Delta.

| Governmental | N | Non-Governmental | N |
|---|---|---|---|
| National Park Regional Directorate | 4 | National NGOs | 2 |
| State Hydraulic Works | 2 | Local NGOs | 4 |
| Ministry of Agriculture and Forestry | 1 | Headmen | 7 |
| Ministry of Urban and Environment Planning | 1 | Company | 1 |
| İzmir Municipality | 3 | Farmer | 1 |
| Karşıyaka Municipality | 1 | Student Club | 1 |
| Menemen Municipality | 1 | | |
| Çiğli Municipality | 1 | | |

### 2.2.3. Field Visits

The fieldwork was conducted from January–June 2021. The fieldwork was designed using the maps and grids developed during the literature review (Figure 1). The field (ca. 80,000 ha) was divided into approximately 200 2 × 2 km UTM grids covering the old and new riverbeds of the Gediz River. The grids were visited with a team of 2 people by car using a transect methodology through each grid. Each transect took approximately 30 min, and the car was travelling at 30 km/h. During the transect, observations were made to determine the presence/absence of the previously identified threats per grid. Then, a simple heat map created with Argics 10.2, was used for estimations building representation of hotspots for increased threats according to their rate in the grid. In addition, threats that were seen but not previously identified in the literature were added to the database. Existing threats that were identified in the literature or interviews but were not visually identifiable were not localized in the maps (such as sea levels increase and other threats); nor were no-longer existing threats (such as cancelled construction projects).

### 2.3. Threat Ranking

Threat ranking with the conservation standards uses criteria-based ranking of threats to provide an objective analysis to determine the importance of each threat. This ranking allows for the identification of critical threats, which are the threats that are the most problematic. First, the threats and conservations suggestions were coded based on the IUCN threats classification system [9,22] and then ranked using the Conservation Standards methodology [11]. We linked the threats to the habitats that they impact (Inland wetlands,

Marine and Coastal wetlands, Agricultural and Grassland habitats, and Mediterranean habitats). The habitat classification is based on the unpublished data of Guelmami [19], 2021, in the hierarchical order given following the bird habitat classification provided in the second edition of the European Breeding Bird Atlas 2 [20,29]. Using the conservation standards threat classification [11], threats were then ranked for each habitat according to four classes: "Very High (71–100%)", "High (31–70%)", "Medium (11–30%)", and "Low (1–10%)" using three criteria: (a) scope (the proportion of the total area affected based on fieldwork observations and literature reviews), (b) severity (based on overall declines caused by the threat according to expert knowledge, importance scores from interviews or literature sources), and (c) irreversibility (based on how long it takes to restore or reverse according to expert knowledge and literature) [11]. The Conservation Standards for the Practice of Conservation and its software platform Miradi were used to rank the threats [30].

## 3. Results

### 3.1. Threats Assessments

#### 3.1.1. Literature Review

A total of 547 news articles were found in Google news; 285 of 547 articles concerned the Gediz Delta; of the 285 news articles, 82 identified threats in the Gediz Delta. A total of 24 scientific publications were evaluated for the specific threats in the Delta [17,24,31–45]. These publications included 17 scientific articles, 4 PhD theses [46–49]; 2 reports [23,50] and one book [51]. A total of 106 sources (scientific and popular journals) were examined, mentioning 233 threats (some of the threats were mentioned multiple times). The threats were coded into 11 classes. The most cited threats were: "residential & commercial development" (20.17%) and "pollution" (19.31%) followed by "transportation & service corridors" (18.45%) and "climate change" (13.73%). The "human intrusions & disturbance" and "energy production & mining" threat categories were also mentioned in the literature review (Figures 2 and 3).

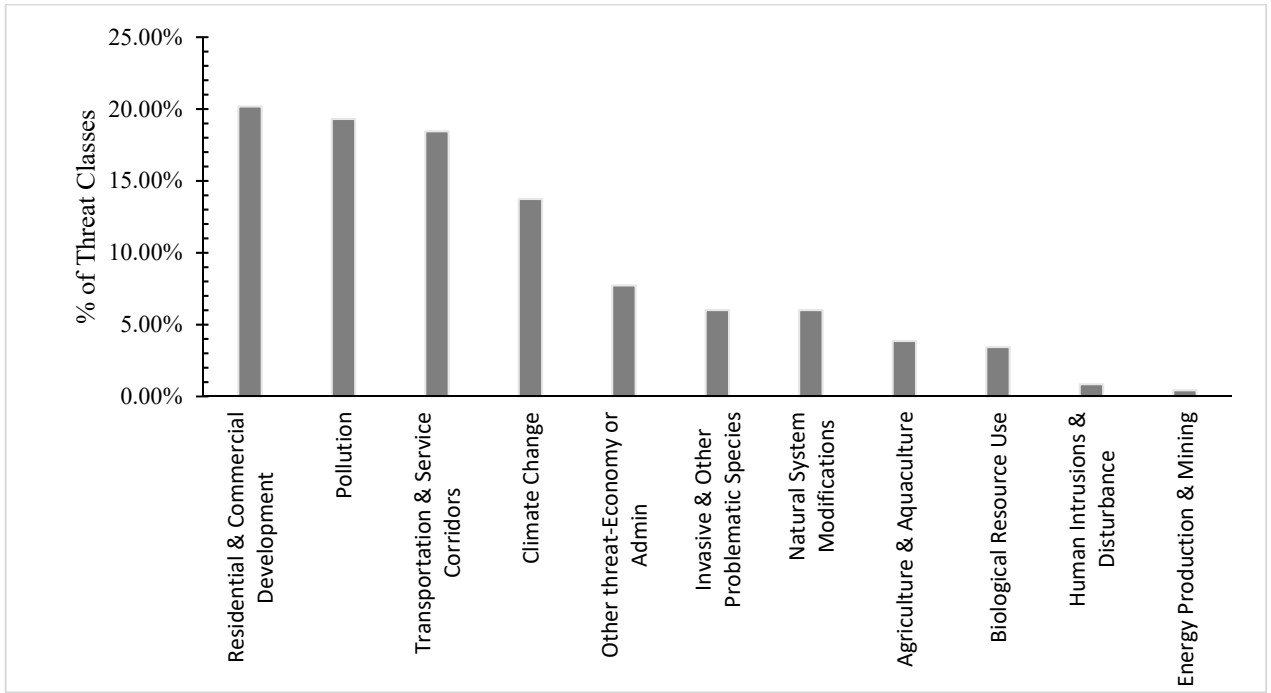

**Figure 2.** Classification of the major threats in the Gediz Delta based on the IUCN threats classification system using a systematic literature review (n = 233 sources).

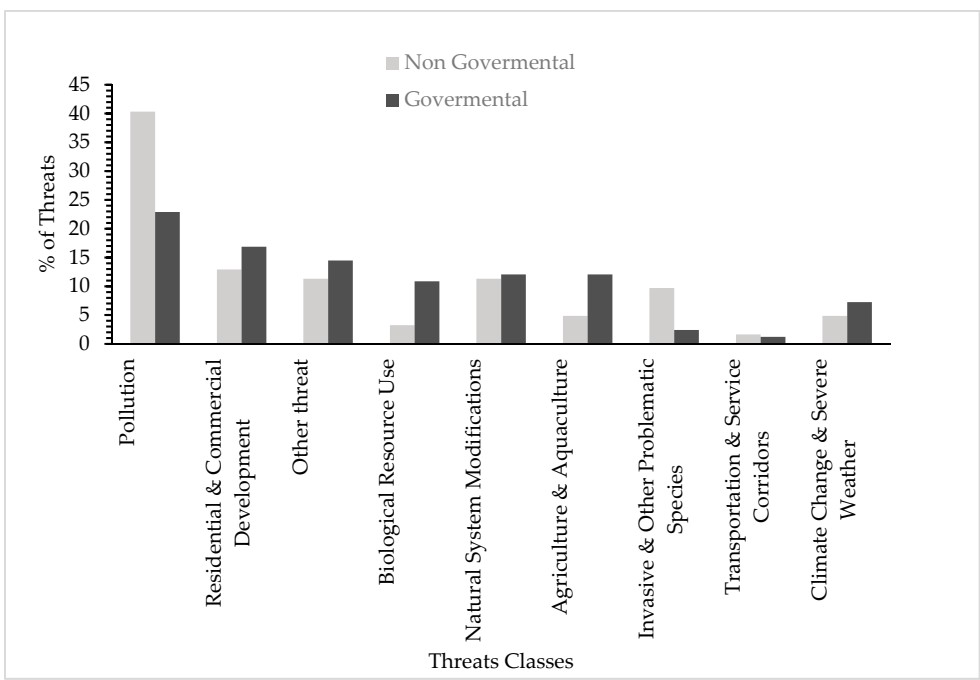

**Figure 3.** Frequency of direct threats identified by governmental and non-governmental stakeholders in the Gediz Delta based on the IUCN threat classification system (30 ind).

### 3.1.2. In-Depth Interviews

To assess the local perspectives, a total of 30 (6 Female, 24 Male) stakeholders from 24 different socio-professional and local groups were interviewed. The stakeholders were from both governmental and non-governmental organizations (14 and 16 people, respectively) (Table 1). A total of 152 threats were identified in the interviews. The threats were coded using the IUCN threats classification system. A total of nine distinct threat categories were identified. The most frequently cited threat in the interviews was "pollution", whereas "urbanization" was most cited in the literature review. Other common threats included "residential & commercial development" (16.45%) followed by "agriculture & aquaculture" (11.84%), and "natural system modification" (11.18%). Pollution was less identified by government stakeholders (21.84%) compared to non-governmental stakeholders (40%). Likewise, "invasive & other problematic species" are less pronounced by the governmental stakeholders (Figure 3). There are significant differences in the responses between governmental and non-governmental stakeholders ($x^2$ = 6.82, df = 1, $p \leq 0.0008$). The most common conservation recommendations included "conservation designation & planning" (21.14%), followed by "awareness raising" (14.63%) and "land/water management activities" (15.45%) (Table 2).

### 3.1.3. Field Visits

A total of 200 grids were visited, and 19 threats grouped into 9 classes were identified. Certain threats such as "human intrusions & disturbance" and "other threats" could not be visually observed in the field visits. The most common threats in the grids were "invasive non-native/alien plants & animals" (observed in 77% of grids), "annual & perennial non-timber crops" (observed in 70% of grids), and "garbage & solid waste" (observed in 67% of grids). The least common threat in the grids is "mining & quarrying" (1%). According to the heat map, the threats are mostly located in the inner part of the Delta, in agricultural & grassland habitats (Figures 1 and 4).

**Table 2.** Suggested conservation actions for the Gediz Delta and their frequency as identified by stakeholders (classified according to the Conservation Standards Methodology).

| Conservation Action Classification | Description | Government Stakeholders | Non-Governmental Stakeholders | Average of Interviews | Examples |
|---|---|---|---|---|---|
| Conservation Designation & Planning | Conservation Planning Easements and Resource Rights Land/Water Use Zoning and Designation Protected Area Designation and/or Acquisition Site Infrastructure | 29.31% | 13.85% | 21.14% | • increase the capacity of the water treatment facilities;<br>• lobby to include the Gediz River into the river boundaries law;<br>• enforce nature and urban planning;<br>• improve intersectoral planning tools;<br>• wetland restoration; |
| Land/Water Management | Ecosystem and Natural Process (Re)Creation Site/Area Stewardship | 10.34% | 20.00% | 15.45% | • intersectoral water resource planning;<br>• reinforce dyke management;<br>• lobby for regular freshwater supply to the Delta; |
| Awareness Raising | Outreach and Communications | 13.79% | 15.38% | 14.63% | • agro-ecological farmer training;<br>• awareness raising campaigns about the Delta's values; |
| Institutional Development | External Organizational Development and Support Financing Conservation Internal Organizational Management and Administration | 17.24% | 6.15% | 11.38% | • increase the number of staff for law enforcement (NGO or Government);<br>• fundraising nature protection activities (such as new water treatment facilities); |
| Livelihood, Economic & Moral Incentives | Linked Enterprises and Alternative Livelihoods Market-Based Incentives | 8.62% | 12.31% | 10.57% | • eco-branding and improved marketing of traditional products; |
| Law Enforcement & Prosecution | Detection and Arrest | 6.90% | 7.69% | 7.32% | • control illegal industrial waste;<br>• removal of illegal livestock yards and fishing houses; |
| Legal & Policy Frameworks | Laws, Regulations, and Codes Policies and Guidelines | 5.17% | 6.15% | 5.69% | • create legislation for organic farming;<br>• lobby for stronger regulation; |
| Research & Monitoring | Basic Research and Status Monitoring | 5.17% | 6.15% | 5.69% | • independent pollution testing and enforcement;<br>• research and monitoring to identify problem species; |
| External capacity building | Alliance and Partnership Development | 0.00% | 9.23% | 4.88% | • collaboration between all stakeholders;<br>• move dog shelter to new location; |
| Species Management | Species Stewardship | 3.45% | 3.08% | 3.25% | • population control hunting;<br>• removal of feral animals;<br>• improved management of the animal shelter;<br>• capture and sterilization campaigns. |

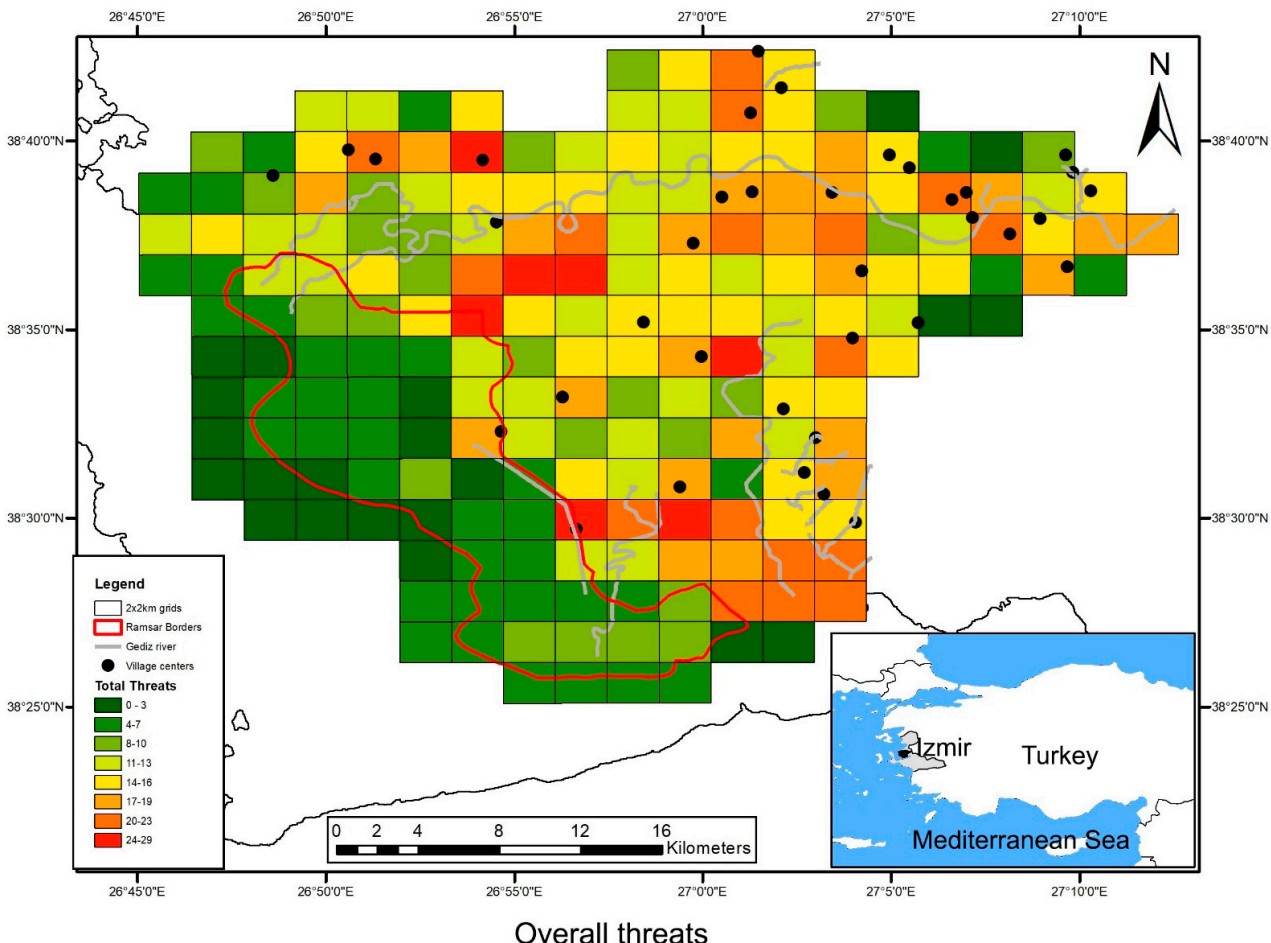

**Figure 4.** Global threat heat map of the Gediz Delta with green representing the lowest threat rankings and red representing highest threat rankings.

### 3.2. Threat Ranking

Overall, 11 threat classes were recorded (Table 3) and evaluated for four key habitats in the Delta. The habitats that were ranked as "very highly threatened" were Inland wetlands, Marine and Coastal Habitats, and Agricultural and Grassland Habitats. Mediterranean habitats are ranked as highly threatened. "Residential and commercial development" and "climate change" are the highest ranked threats and have the potential to impact all habitats in the Delta, followed by "transportation/service corridors" and "pollution". The category of "residential & commercial development" includes (1) "commercial & industrial areas", (2) "housing & urban areas", and (3) "tourism & recreation areas" in the Delta. The "climate change" threat includes "droughts" and "habitat shifting & alteration" across the Delta. The "transportation services and corridors" threats are mainly evaluated as an existing threat with utility services (such as electric poles) and flight paths (on the southern coast; there is a military airport). The third direct threat in the Delta is pollution from (1) "agricultural & forestry effluents", (2) "garbage & solid waste", (3) "household sewage & urban wastewater", and (4) "industrial & military effluents". The category of "other threats" (such as management problems) and "human intrusions" were evaluated as directly affecting only inland and coastal marine habitats with low impact. "Energy production & mining" threats were indicated for Mediterranean habitats (Table 3). The threat ranking combined with the heat maps shows that the most threatened habitats are in the inner part of the Delta (agricultural & grassland habitats) (Figure 4 and Appendix A). The inner parts of the Delta are mainly threatened by "pollution" and "residential and

commercial development" (Table 3), whereas the marine and coastal areas are threatened by "climate change".

**Table 3.** Threat impacts ranking of key habitats of the Gediz Delta using the Conservation Standards Methodology. Threat impacts have been categorized as Low (Dark Green), Medium (Light Green), High (Yellow), Very High (Red), and not existing (White).

| Threats\Targets | Inland Wetlands | Marine Coastal | Agricultural Grassland | Mediterrean Habitats | Summary Threat Rating |
|---|---|---|---|---|---|
| Residential and Commercial Development | Very High | Very High | Very High | Very High | Very High |
| Climate Change | Very High | Very High | Medium | Medium | Very High |
| Transportation and Service Corridors | High | Very High | Medium | Medium | High |
| Pollution | High | High | Very High | Low | High |
| Invasive and Other Problematic Species | Medium | Medium | Medium | Medium | Medium |
| Agriculture and Aquaculture | Medium | Low | Medium | Medium | Medium |
| Natural System Modifications | High | Medium | Medium | Low | Medium |
| Biological Resource Use | Medium | Low | Low | Low | Low |
| Other threat-Economy or Admin | Low | Low | | | Low |
| Human Intrusions and Disturbance | Low | Low | | | Low |
| Energy Production and Mining | | | | Low | Low |
| Summary | Very High | Very High | Very High | High | Very High |

## 4. Discussion

A systematic analysis of the threats in the Gediz Delta using a multi-method approach allowed us to identify a wide panorama of threats that are both perceived and observed. Similar to other studies concerning the threats to wetlands around the world [6,12], "residential & commercial development" and "climate change" are ranked as very high direct threats contributing to habitat destruction. The main driver of "residential & commercial development" is urbanization; this threat was only minimally evoked in the stakeholder interviews, yet it was quite important in the literature review and field survey (the built-up area increased by 85% over 40 years [24]). The reason that urbanization is less mentioned in the interviews could in part be due to the fact that the construction occurred outside the protected area, and these areas are not seen as a threat to the natural areas. Most of the urbanization also occurred before the 2000s [24], which could make this threat be seen as a past threat rather than a current threat. Despite this fact, it is important to highlight that the literature review continues to show the ongoing construction proposals for the Delta, indicating a continued threat [18].

Climate change was ranked as a very high threat, affecting all habitats in the Delta. The direct impacts of climate change could only be partially observed (sea-level increases and droughts), yet the projected scenarios for the region indicate serious risks in the future [8,41]. Climate change was not commonly evoked by the stakeholders (less than 10%) nor in the literature (13.73%); however, the reoccurring droughts between 1988 and 1997 caused severe drying of the wetland and grassland habitats. The climate change projections forecast more severe droughts in time, which will adversely affect the populations of many species living in the Delta [24]. Additionally, the current data indicates that climate change is impacting and will continue to impact the marine and coastal wetlands, especially lagoons [41,52,53]. In order to reduce the effects of these threats, some restoration work was carried out in the Delta. First, in 1999, freshwater was pumped to prevent inland marshes from drying up [24,51] and second, the flamingo breeding islands that were destroyed by waves were restored in the salinas [52]. These restoration activities are only short-term solutions and are necessary to repeat over time. In order to protect these valuable habitats, new and alternative solutions must be found using a more global approach [8,54]. However, short-term solutions are critical for the sustainability of these habitats in the Delta, especially for

inland wetlands that are directly dependent on continual freshwater sources. In order to maintain these resources, continued lobbying for regular freshwater input into the Delta is highly recommended. This implies a collaboration between the environmental and agricultural sectors to ensure a sustainable balance of water use [54].

The "transportation service and corridors" threat in the Delta is often associated with the threat of "residential & commercial development" in both the literature and stakeholder interviews. According to the threat heat map, the two threats were concentrated in the inner part and on the periphery of the Delta; however, the coastal parts of the Delta have been threatened with projects that did not materialize. These planned, or announced but cancelled projects include building a container port, public beaches, an amusement park (Disneyland), a new fairground project, a mega-bridge construction project and a skyscraper project in Mavişehir (on the Delta shoreline) [51,55]. The fact that the Delta is located next to the city of İzmir causes an increase in demand for urban development in the region [18,49]. This threat can be seen around the Mediterranean basin with increasing human population pressures causing the transformation of 45–51% of natural wetlands into agricultural and urban zones since 1970 [6,54]. The protection status of Gediz Delta has had a significant impact, preventing many construction projects on the coastline [51,56]; however, it has been shown that legislation alone is not sufficient [54]. To reduce the threats of "transportation services and corridors" and "urbanization", it is imperative to increase social awareness about the protection and sustainable management of Mediterranean wetlands and to improve intersectoral planning and collaboration [57].

The "pollution" threat is most prominent in the stakeholder interviews. The perception of this threat could be as it is often more visible than other threats (illegal dumps) in the Delta, and its impact is felt directly by the local population. "Pollution" was directly observed in more than 60% of the grids. All three sources of information (literature review, stakeholder interviews, and field visits) have identified the Gediz River as a primary source of pollution in the Delta, and it has been cited as one of the most heavily polluted rivers in Turkey due to agricultural drainage water, industrial wastewater and domestic wastewater [34,37,40]. This pollution is enhanced by 400 leather factories in Uşak and 57 leather, oil, and soap factories in Manisa [58]. The change in farming practices, with extensive vegetable and fruit production being replaced with intensive cotton and vegetable farming, has contributed to increased pollution [36]. One participant in the interview commented that "the river's water was directly drinkable by locals forty years ago, but after the 2000s, the increased pollution in the river has had adverse effects in agriculture productivity and quality". There have also been reports in the press about mass fish deaths along the river [59,60]. In addition to water pollution, the threat of garbage and solid waste problems, such as plastics, domestic waste, and rubbles, reduces the quality of the Delta's habitats. Garbage and solid waste problems were found in 67% of the grids. Pollution is an important variable that threatens not only the Gediz Delta, but also 54% of the world's wetlands [12]. The EU Water Framework Directive (2000/60/EC) policies have positively reduced pollution and nutrient concentrations in surface water in European countries [54]. In this perspective, we highly recommend that the Gediz River be included into the river boundaries law in Turkey; this will create the necessary legislation to enforce water quality in the Delta. Secondly, it is necessary to increase the capacity of water treatment facilities to reduce overflow and direct discharges into the river. Another management recommendation is to promote organic and sustainable farming practices to reduce pesticides and other agricultural inputs from entering into the water supply [61].

The threats in the Delta that were ranked as medium include "invasive and other problematic species", "agriculture and aquaculture", and "natural system modifications". All these threats were identified in the literature review, interviews, and field observations, yet the frequency and importance given to the threats varied significantly depending on the method used. "Invasive and other problematic species" in the Delta include eucalyptus trees, feral horses, dogs and cats, and wild boars. These problematic species were observed in 77% of the grids. The feral cats and dogs prey on many wild bird and animal

species [62,63], impacting the biodiversity. An example of this impact can be seen in the Delta when feral dogs attacked the flamingo nests repeatedly from 2006–2011, breaking thousands of eggs (3600 eggs in 2011 and 2189 eggs in 2010) [51]. Feral dog collection campaigns and barbed wire fencing were used to tackle these problems, and we recommend that these strategies be maintained to control the feral dog population in the future. On the other hand, feral horses are categorized as a threat, yet they also hold an ecological role for maintaining the habitats open in the absence of other wild grazing animals [50,64]. This indicates that population size should be controlled through management practices, but eradication should not be promoted as this could have even greater consequences on the biodiversity of the Delta. Another example of the "invasive and problematic species" threat are the eucalyptus tree plantations and their impact in terms of water demands [18]. The trees were initially planted in the Gediz Delta in the 1970s to open residential and recreational areas, based on the idea that the trees would help dry the marshes that were considered a source of malaria [18]. It should be noted here that these trees are no longer planted today, and they are gradually being removed and replaced by native tree species overtime [50].

The "agriculture and aquaculture" threat mainly focused on agricultural intensification (for both agricultural crops and livestock farming) [23,32,36,45,47,50,51]. This threat was observed in 70% of the grids. Despite this intense scope, the threat was evaluated as having a medium impact as the potential for irreversibility. In order to reduce this threat, two conservation activities are recommended: (1) agro-ecological farmer training and (2) branding and improved marketing of traditional products. Promoting this kind of activity could bring mutual benefits for both biodiversity and landowners [65].

The "natural system modification" threat category was also evaluated as "medium" severity in the threat ranking; however, it is a high threat for inland wetlands due to river bed infrastructure and increased channels that reduce freshwater sources in the Delta [24]. In addition, the threat was commonly expressed by stakeholders with inefficient freshwater circulation in the channels and salt pans and negative impacts of the dams on the river and the water regime. Despite this concern from the stakeholders, there was no evidence in the literature showing impacts of channelization, saltpan extension, or dams in the water circulation of the Delta. To better understand this discrepancy, we recommend implementing an improved water monitoring schemes to apprehend the water circulation in the Delta and create a management system to use the existing water more effectively. These data could provide the necessary information to put in place a restoration project in part of the abandoned Salinas, improving water circulation and the development of dunes and temporal coastal wetlands in the Delta [66].

The threats that were ranked as low include "biological resource use", "other threat-economy or administration", "human intrusions & disturbance", and "energy production & mining". The literature review identified various "biological resource use" threats including poaching, improper reed management, overfishing, and grazing activities [47,50,53,67]. These threats were often associated with lack of regulation or limited policy control due to insufficient staff. Although the "biological resource use" threat is not commonly cited in the Delta, and it was evaluated with a low impact severity, it should be noted that this threat is one of the four most common threats in wetlands worldwide [7,12]. Thus, this study might have underestimated the threat severity due to limited literature and low attention given by the stakeholders. To reduce the impacts of this threat, we recommend further research and monitoring studies be implemented to identify impacts of the threat both on biodiversity and ecosystem services. Based on this information, it can be determined if there is a need to develop new laws or regulations and/or enforce staff (NGOs or Government) to ensure the implementation of existing laws.

The threats "human intrusions & disturbance" and "energy production & mining" were not identified by the stakeholders and were only vaguely mentioned in the literature. There were some articles that mentioned concern about noise disturbances from military airports [46,47]; however, this threat was not evaluated, as it could not be precisely localized

in the grids. "Energy production and mining" was observed with two indirect threats: "mining and quarrying" and "renewable energy". This threat was observed with solar energy farming, but the other references from the literature were unrealized projects [68]. These planned projects could pose eventual threats in the Delta in the future and attention should be given to avoid the impact that they could have on the biodiversity.

The overall threat analysis ranked "coastal and marine", "inland wetlands", and "agricultural & grassland" habitats as highly threatened in the Delta. On the contrary, the heat maps do not show this ranking, as some of the important threats were not visible during the field visits such as planned but unrealized projects and sea level rise. Given that the Delta is protected by national laws, some construction projects proposed for coastal and marine habitats were abandoned [51], yet despite this protection other projects have caused serious loss to inland wetlands in last decades [17,24,51,69] with irreversible loss to some critical habitats (such as the Çiğli marches) [24]. It is expected that many of the exiting threats will continue due to the geographical proximity of the Delta to the city of İzmir, with increasing urbanization pressures [18,46] and climate change projections. This highlights the need to conserve habitats in the protected areas and to target the conservation activities that will reduce the threats affecting these habitats.

Unfortunately, the agricultural and grassland habitats are often outside of the highly protected areas, yet in our study these are the habitats that are the most threatened. It has been shown that agricultural and grassland habitats surrounding wetland ecosystems have an important role in conserving wetlands as they provide feeding and breeding habitats for many species [70,71]. Therefore, it is essential that conservation activities include all of the Delta's terrestrial ecosystems, especially those largely forgotten in conservation policies [20,71].

The use of different methods and perspectives to identify threats in the Gediz Delta allowed us to pinpoint the critical threats in the Delta. This multi-method approach is useful to better understand where consevation efforts could and should be undertaken. The threats mentioned in this study are common to many wetland ecosystems around the world [12]. However, it is important that they be evalauted in each specific context. Taking into account different perspectives (stakeholders' perceptions, media, and scientific research) can contribute to the success level of the conservation strategies [72–74].

## 5. Conclusions

This study identified both the perceived and observed threats impacting the habitats, and its biodiversity in the Gediz Delta using a multi-method approach involving local and scientific knowledge systems. The threat ranking could have had some bias, given that it was based on expert evaluation, but this bias was reduced with the inclusion of literature and survey results. The difference in results from each collection method shows the importance of using a multi-method approach to understand the dimensions of the threats fully. This methodology can be applied in other wetlands to prioritize the threats and understand the cumulative effects on both habitats and species. The threat analysis in the Gediz Delta shows the importance of enlarging conservation activities outside of the strictly protected. This analysis also provides the grounds to identify the most appropriate conservation strategies that could be applied to the site in order to have the most impact.

**Author Contributions:** D.A. was responsible for the conceptualization, data collection and analysis, creation of the figures and tables, and drafting and coordinating the drafts of the article. L.E. was implicated in the conceptualization, data analysis, and drafting and editing the drafts of the article. She was also responible for project adminstration and funding. K.Ç. was implicated in the data analysis, creation of figures and tables, and editing the drafts of the article. He was also responsible for the ethical and field authorisations. Ö.D. was implicated in the data analysis, creation of figures and tables, and editing the drafts of the article. All authors have read and agreed to the published version of the manuscript.

**Funding:** This study was funded by the Foundation Tour du Valat and the Campus France Scholarship.

**Institutional Review Board Statement:** The study was conducted according to the guidelines of the Declaration of Helsinki, and approved by the Institutional Review Board of Aegean University of Izmir.

**Informed Consent Statement:** Informed consent was obtained from all subjects involved in the study.

**Data Availability Statement:** Raw data were generated at the Tour du Valat. Derived data supporting the findings of this study are available from the corresponding author on request.

**Acknowledgments:** We acknowledge the volunteers and professionals involved in interviews. We also thank to Esra Kartal for her help in accessing key stakeholders, and Juliette Biquet and Pelayo Menéndez Álvarez for their help during the field work. We would also like to thank Anis Guelmami for his help with the maps.

**Conflicts of Interest:** The authors declare that there are no conflicts of interest on any financial or non-financial (political, personal, professional) interests/relationships that may be interpreted to have influenced the manuscript.

## Appendix A

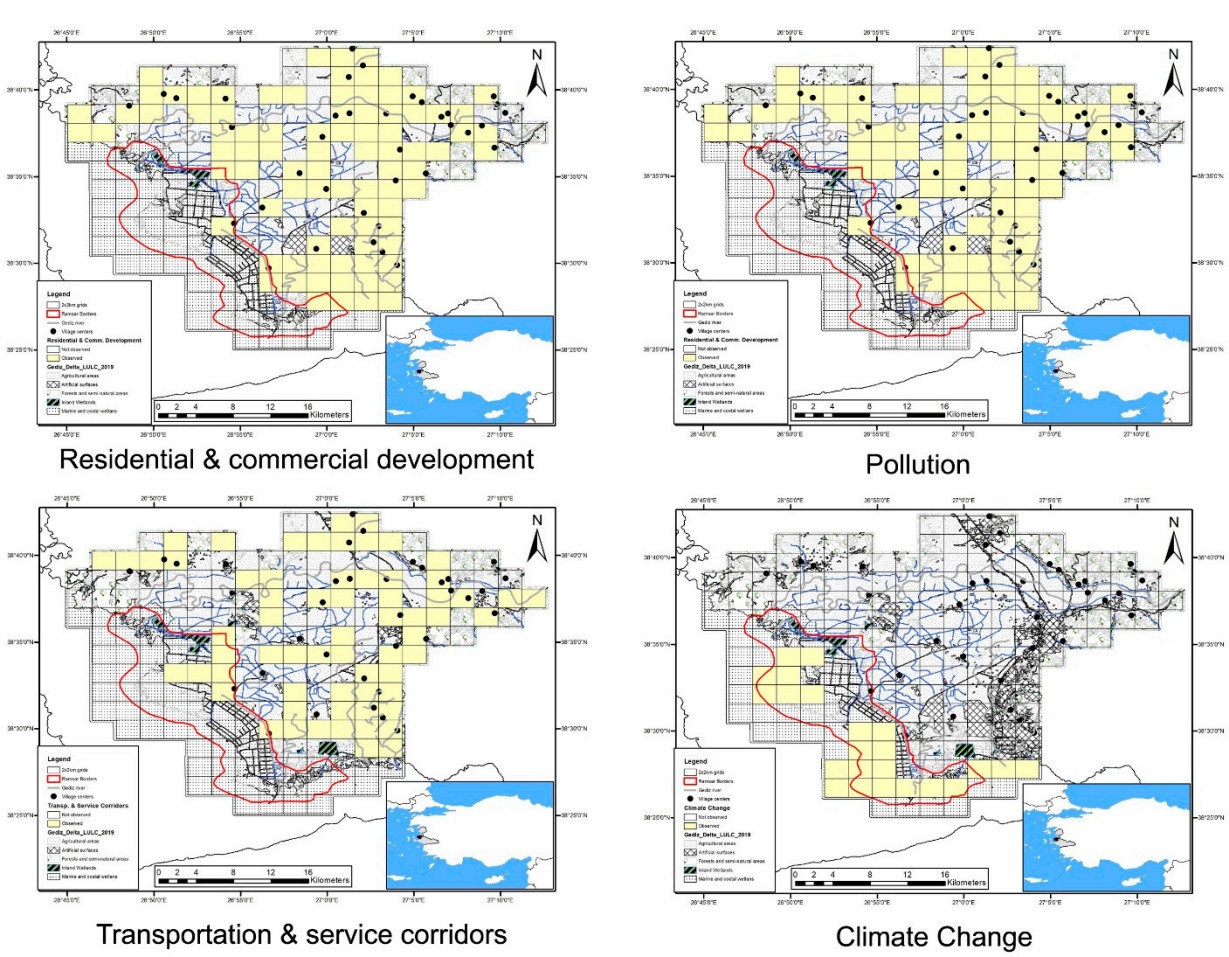

**Figure A1.** The maps show the grids where the four major the threats were identified through the field survey. Yellow colour represents where the threat was observed.

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
