# Peer review of "Threat Ranking to Improve Conservation Planning: An Example from the Gediz Delta, Turkey"

_land, doi:10.3390/land10121381_

Round 1

Reviewer 1 Report

Authors assessed the threats to a wetland of international importance using a multi-method approach. This is a very interesting and useful study that would be also useful as an example for other similar areas. It could be recommended for publication after some improvement.

Journal style has not been fully implemented (e.g., line spacing, figure and table positioning, in-text references).

Specific comments

---Line 14 – Mediterranean wetlands are among the most…

---Line 24 – the most important threats are…

---Line 25 - a need to extend…

---Lines 41-42 – Give reference.

---Line 47 – will change and destroy…

---Lines 78-79 – Authors explained this later, but a few words are also needed here for clarity, e.g., indicate some structural changes in parentheses.

---Lines 133-135 – Not clear. Revise text.

---Lines 158-160 – Define threat classes according to Conservation Standards here. It would be helpful for the reader to better understand methods and results. It would be inconvenient to search for the methodology.

Author Response

Responses to The First Reviewer

Reviewer 1: Thank you for your editing suggestions.  We have taken into account all of the suggestions:

---Line 14 – Mediterranean wetlands are among the most…

---Line 24 – the most important threats are…

---Line 25 - a need to extend…

---Line 47 – will change and destroy…

---Lines 41-42 – Give reference.

Response: We cited “Geijzendorffer, I.; Chazée, L.; Gaget, E.; Galewski, T.; Guelmami, A.; Perennou, C.; Davidson, N.; McInnes, R. Mediterranean Wetlands Outlook 2: Solutions for Sustainable Mediterranean Wetlands; Secretariat of the Ramsar Convention, 2018” and “Galewski, T.; Segura, L.; Biquet, J.; Saccon, E.; Boutry, N. Living Mediterranean Report – Monitoring Species Trends to Secure One of the Major Biodiversity Hotspots; Tour du Valat: Arles (FRA), 2021; p 20.”

---Lines 78-79 – Authors explained this later, but a few words are also needed here for clarity, e.g., indicate some structural changes in parentheses.

Response: We revised the explanation in line 83

---Lines 133-135 – Not clear. Revise text.

Response: We revised the text in lines 133-135

---Lines 158-160 – Define threat classes according to Conservation Standards here. It would be helpful for the reader to better understand methods and results. It would be inconvenient to search for the methodology.

Reviewer 2 Report

First of all, congratulations for the work.

I found interesting the way of testing the problems using the news and scientific studies and comparing them.

The document is well presented and explained, I only have a few small questions.

L180- a total of 30 (3 Female, 23Male) = 26.

Figure 5 appears to be very large and is reduced in the text.
As an idea, I suggest that you leave the central image as figure 5 and place the images on the sides in an appendix.

With these small changes I think it would be ready for publication. 

Author Response

Responses to The Second Reviewer

Reviewer 2: Thank you for your attention to these details!  We have taken into account your recommendations.  Specifically we have modified the following:

L180- a total of 30 (3 Female, 23Male) = 26.

Response: We corrected the numbers to a total of 30 (6 Females, 24 Male).

Figure 5 appears to be very large and is reduced in the text.
As an idea, I suggest that you leave the central image as figure 5 and place the images on the sides in an appendix.

Response: Thank you for this comment. We updated the maps based on the comment: with one map (fig 4) in the text and the others included in the supplementary material.

Reviewer 3 Report

This is an interesting paper discussing the need to extend conservation actions in the inner part of the Gediz delta. The manuscript is overall well written but needs minor revisions, as follows:

In the introduction the authors should clearly provide the aim/scope and hypotheses of the paper at the end of this section.

Figure 2 should be revised to be able to read the variables.

From my perspective, Figure 3 should be removed – creating a word cloud is not a representative scientific method in a research study. The authors should better summarize these findings in a phrase.

In the discussion section, the author should better indicate the importance of stakeholders in conservation practices and refer to international studies (i.e.  Rozylowicz L. et al. (2017), Recipe for success: A network perspective of partnership in nature conservation. Journal for Nature Conservation 38: 21-29; Manolache S., et al. (2018) Power, influence and structure in Natura 2000 governance networks. Journal of Environmental Management, 212: 54–64; Bodin, Ö. et al., 2016. Collaborative Networks for Effective Ecosystem-Based Management: A Set of Working Hypotheses. Policy Studies Journal.)

The authors should better highlight the contributions and extensions of their approach at international level.

Author Response

Responses to The Third Reviewer

Thank you very much for your ideas.

In the introduction the authors should clearly provide the aim/scope and hypotheses of the paper at the end of this section.

Response: Thank you for this comment. We updated our aim/scope in 81-83 lines.

Figure 2 should be revised to be able to read the variables.

Response: Thank you for this comment. We corrected figure 2, and now figure 2 can be readable.

From my perspective, Figure 3 should be removed – creating a word cloud is not a representative scientific method in a research study. The authors should better summarize these findings in a phrase.

Response: Thank you for this comment. We removed Figure 3 and summarized the findings in lines 192-193.

In the discussion section, the author should better indicate the importance of stakeholders in conservation practices and refer to international studies (i.e.  Rozylowicz L. et al. (2017), Recipe for success: A network perspective of partnership in nature conservation. Journal for Nature Conservation 38: 21-29; Manolache S., et al. (2018) Power, influence and structure in Natura 2000 governance networks. Journal of Environmental Management, 212: 54–64; Bodin, Ö. et al., 2016. Collaborative Networks for Effective Ecosystem-Based Management: A Set of Working Hypotheses. Policy Studies Journal.)

The authors should better highlight the contributions and extensions of their approach at international level.

Response: Thank you for these comments and the suggestions for the references.  We found the recommended articles interesting and useful. We wrote a new paragraph in lines 439-447 to discuss the stakeholders' importance and highlight the contributions with the citation of these 3 articles.